# Dialogue, Worldview Inclusivity, and Intra-Religious Diversity: Addressing Diversity through Religious Education in the Finnish Basic Education Curriculum

**Martin Ubani \*, Elisa Hyvärinen, Jenni Lemettinen and Elina Hirvonen** 

School of Applied Educational Science and Teacher Education, Philosophical Faculty, University of Eastern Finland, 80101 Joensuu, Finland; elisa.hyvarinen@uef.fi (E.H.); jennilem@uef.fi (J.L.); elinhi@uef.fi (E.H.)
\* Correspondence: martin.ubani@uef.fi

**Abstract:** The purpose of this article is to discuss how religious and non-religious diversity are addressed in the current national core curriculum for religious education (RE) in basic education in Finland. We first discuss the educational developments behind the Finnish curricular reform, and then focus on issues related to RE and RE research in Finland. We then describe the key contextual contributors to the current RE curriculum in basic education before proceeding to examine how diversity is addressed in the curriculum. Based on our examination, we identify four themes in the curriculum: inter-religious diversity, religious and non-religious worldviews, cultural diversity, and dialogue skills. In RE, diversity is largely addressed within a framework of religion and multiculturality. The article ends with a call for renewal of research foci in RE.

**Keywords:** religious education; diversity; multiculturalism; curriculum; public education

## 1. Introduction

The aim of this article is to discuss how religious and non-religious diversity is addressed in Finnish religious education (RE) in the current National Core Curriculum for Basic Education (NCCBE 2014). The Finnish basic education consists of grades 1 to 9 and covers ages 7 to 16. The Finnish educational system is similar to other Nordic countries: nearly all Finnish children attend state-funded public schooling. There are almost 2200 elementary schools (Official Statistics of Finland OSF). Under 100 of those are private schools and only under 20 are schools with some religious devotion. At grades 1–6, RE is taught by class teachers and at grades 7–9 teaching is provided by subject teachers specialized in RE.

The underlying position of this article is the view that recognition of diversities in public education is essential not only for Human Rights purposes but also in order to equip the citizens of today and tomorrow with sufficient skills required for navigating in evolving plural and multicultural societies of the 21st century. In our examination, we use the concept "diversity" in a heuristic manner, referring to the inclusion of a variety of people into a determined group (see Merriam-Webster 2020; Cambridge English Dictionary 2020). While we acknowledge the broad discussion on the concept of diversity and its developments (Quinn et al. 2018; Driedger 2008; Doucerain et al. 2013) and the historic layers present in each curriculum, for the purpose of this article, it is sufficient to approach the topic mainly inductively.

In this article, we first discuss the educational developments behind Finnish curricular reform, then focus on issues related to RE and RE research in Finland. Drawing on this, we then describe the key contextual contributors to the current RE curriculum in basic education (NCCBE 2014) before proceeding to examine how diversity and religious and non-religious diversity in particular are

presented in the curriculum. In Finland, all instruction in public schools is subject to the National Core Curriculum for Basic Education issued by the Finnish Board of Education every 10 years. In it over 450 pages, it includes objectives and core contents of each school subject, including RE for grades 1–9 (from children aged 7 years onward).

In the literature on religion in public education, the demand for pluralization and diversification of societies in Europe has been consistently acknowledged in recent decades (Meijer et al. 2009; ter Avest et al. 2009; Jackson 2014; Ubani et al. 2019a). In Finnish society, too, the processes of secularization and pluralization have been increasingly recognized in the research on public education, although the pace of these developments has been somewhat slower than some of the more multicultural and pluralistic countries of Europe (Ubani et al. 2019b). The new National Core Curriculum for Basic Education, issued in 2014 and which came into effect in 2016, reflects the recognized need to address multiculturalism, inter-cultural skills and diversity, which have been at the forefront of recent curricular and educational thinking (see Zilliacus et al. 2017; Räsänen et al. 2018; Ubani 2013). The issues advocated in the curriculum, such as incorporating local diversity in the school community (NCCBE 2014), are also of relevance to RE, as, for instance, religious and non-religious traditions in local communities are to be recognized in RE instruction more prominently than before (NCCBE 2014; NCCBE 2004). However, there are major differences in religious demography within Finland. This partly makes it possible, along with the skills-based approach, to have stronger local differentiation in the content of instruction than previously in RE in basic education (NCCBE 2014).

For nearly a decade, the most active and pressing topic related to RE in Finland has been the question of integrated education. Arguably, the aforementioned differences in religious demography are also related to the varying degrees of pressure toward integrative practices in RE. In Finland, the provision of RE is segregated according to the pupil's own religious affiliation (NCCBE 2014; Ubani and Tirri 2014). However, for example, in Eastern Finland some elementary schools provide nothing other than Lutheran RE or Orthodox RE and ethics lessons are very poorly attended, whereas in Southern Finland some schools provide education in three or more other religions in addition to the three denominations mentioned above. Initiatives toward integrated instruction have, indeed, emerged predominantly from Southern Finland and more urbanized areas. For instance, the first fully integrative Worldview Education (WE) experiment took place at the European School of Helsinki (Korkeakoski and Ubani 2018). This was possible due to the European School having an independent curriculum, which enabled the incorporation of new school subjects in their instruction (Kallioniemi 2013), whereas other schools were, and still are, obligated to follow the national Basic Education Act.

Arguably, the several initiatives carried out by municipalities toward integrated instruction have been grounded on the needs of the constantly pluralizing and secularizing environment (Åhs et al. 2016; Kimanen 2016c). These initiatives can be seen as an attempt to respond to the pressure with regard to dialogue: since the early 2000s there has been a recognition that the Finnish model of RE lacks opportunities for dialogue between pupils from different religious or non-religious worldview backgrounds (Ubani and Tirri 2014; Kallioniemi and Siitonen 2003; Franken 2017). Integrated instruction could answer the practical challenges of scheduling segregated instruction with its various forms of RE (Kallioniemi and Siitonen 2003). However, this issue is not without controversy, and several integrated initiatives have led to appeals to different Finnish authorities. For instance, appeals have questioned whether integrated instruction can fulfil the aims and contents of the curriculum of RE and ethics (National Ombudsman of Finland 2017).

## 2. Educational Background to the 2014 Curriculum Reform in Finland

Finland is known for its equalitarian educational system and high-quality teacher education. In Finland, teachers are trusted and given a great deal of professional freedom, which also makes the profession very demanding (Niemi et al. 2016, 2018). Despite this considerable freedom, the curriculum

remains the central working tool for Finnish teachers in planning and developing their own practice (Halinen 2018).

Unlike many other countries, Finland has an open and collaborative system for designing the national curriculum. The national core curriculum is reformed approximately every decade. The process of devising the curriculum aims at being open and transparent; it is an outcome of a broad and inclusive national discussion. The reform is led by the Finnish National Board of Education (FNBE), a governmental development agency responsible for planning, organizing, implementing, and promoting the core curriculum reform with stakeholder participation. Core curriculum work involves, for instance, officials, researchers, educational professionals, and textbook publishers. For example, in the case of RE, the parties involved in commenting on the draft included the Finnish Parent's League, religious communities, and the secular Union of Freethinkers. Schools, teachers, and citizens can also comment on the drafts via the internet (Zilliacus et al. 2017; Tikkanen et al. 2017; Halinen 2018).

In recent years, the Finnish educational system has implemented a significant national core curriculum reform for basic education and high schools (Halinen 2018). Since the late 1900s, experts in Western educational policy and research have discussed how to face the needs of a rapidly changing world. The basis of learning in the latest curriculum is grounded on the pupils' role as active participants (NCCBE 2014). As in most Western countries, the new Finnish public curriculum adopted a skills-based approach (Niemi et al. 2018; Uljens and Rajakaltio 2017; Palsa and Mertala 2019). Discussions about education in this context often involve concepts such as "transferable skills," "new learning skills," and "21st century skills" (Darling-Hammond 2006; Newton and Newton 2014; Viinikka et al. 2019).

The main goal of the reform in Finland was to create better conditions for successful teaching and to enable pupils to develop their own lifelong learning skills and prepare them to maintain a sustainable lifestyle for the future (Airaksinen et al. 2016; Halinen 2018; Viinikka et al. 2019). One main source of discussion on the reform has been so-called phenomenon-based project studies where pupils work in collaboration with other classrooms and with several teachers on topics crossing subject boundaries. As a result, pupils are required to participate each year in at least one such multidisciplinary learning module. In addition, the discussion of phenomenon-based learning seems to have led to some misunderstanding that the nature of the new curriculum has fundamentally changed. However, the national core curricula are still subject-based, but in order to meet the challenges of the future, collaborative methods are also emphasized (Niemi et al. 2018).

In the latest curricular reform, seven transversal competency areas, described as a combination of knowledge, skills, values, attitudes, and will, were introduced to Finnish public education. Transversal competencies of pupils are to be accordingly developed in every subject through both content and methods typical of the discipline in question (Palsa and Mertala 2019). The seven competency areas defined are: (1) Thinking and learning to learn, (2) Cultural competence, interaction and self-expression, (3) Taking care of oneself and managing daily life, (4) Multiliteracy, (5) Information and communication technology competence, (6) Working life competence and entrepreneurship, and (7) Participation, involvement and building a sustainable future (NCCBE 2014).

In line with the new educational thinking of the new curriculum, teachers' work is not limited to the classroom; their responsibility is to work with a wide variety of societal partners. The new curriculum emphasizes that learning environments reflect the global world in which children live. That world is full of digital tools, media services, and games. Moreover, it emphasizes that pupils should be guided and encouraged toward independent, critical searching and use of information (Niemi et al. 2018). In addition, the curriculum for RE aimed at being progressive in its approach by perceiving religion as a positive phenomenon but with negative aspects—including developing skills for critical thinking with regard to one's own tradition, dialogue, and recognition of non-religious traditions and intra-religious plurality, and with a life skills emphasis.

### 3. The Segregated Model of Religious Education as a Context for Curriculum Reform

The Finnish model of RE and Ethics is distinctive compared to many other systems of RE in public education. For instance, compared to other Nordic countries, Finland is the only country where pupils are divided into separate classes based on their religious affiliation (Osbeck and Buchardt 2017; Hartman 2017; Rothgangel et al. 2014). Since the early 2000s, religious education in Finland has been given in a segregated manner according to "one's own religion" (NCCBE 2004, 2014). This is grounded on a positive interpretation of freedom of religion and human rights (Ubani and Tirri 2014; Ubani 2019), according to which pupils are entitled to religious education that is based on their religious affiliation as part of public education. Since the 2000s, as many as 13 religions or religious communities have been represented in the national core curriculum, in addition to which Ethics instruction has been provided to pupils not affiliated with any religion (Ubani and Tirri 2014; Kallioniemi and Ubani 2016). The instruction has been described as "weak confessional" (Ubani 2007; 2019) as, despite separate instruction groups and a content emphasis based on respective religions, information on other religions and worldviews, the instruction includes no faith formation or devotional aims. Furthermore, the qualification of RE teachers is solely academic, with no requirement for membership of a religious community. Research on state policies and educational policies has shown that justification of the current segregated model has been based on contributing to multiculturalist integration, recognition of diversity, state security, and social harmony (Ubani 2013; Sakaranaho 2013).

About 85 percent of eighth graders (age 14–15) take part in Lutheran RE (Ubani 2019; Saarelainen 2018), which exceeds the percentage of the population that are members of the Lutheran Church. At the same time, demand for ethics education has been growing. This is largely due to the growing number of people leaving the Lutheran Church, which has been, and still is, the largest religious community in Finland. The majority of those leaving the Church remain unaffiliated to any religious community (Ubani et al. 2019b). Finland has 5.53 million inhabitants. Estimates indicate that membership of the Finnish Evangelical Lutheran Church will continue to decrease, reaching below 50% of the population by 2040 (Evangelical Lutheran Church of Finland 2019). Even though the majority of the Finnish population still belongs to the Lutheran Church, there is a vast diversity of religions and beliefs in Finland. The official statistics on membership of different religions and denominations are based on membership of registered religious communities. As not all followers of different traditions are counted in the official statistics, the estimations of adherents vary. The largest religious minorities are the Orthodox Church with approximately 62,000 followers and Muslims with, in 2015, approximately 13,000 registered members in several registered communities (Official Statistics of Finland OSF), although the qualified estimate of the total number of Muslims is between 65,000–75,000. There is also diversity within the Lutheran Church. For example, one of the revivalist movements, the Conservative Laestadian revivalist movement, is estimated to have 80,000–100,000 followers.

Arguably, the character of RE instruction, especially Lutheran RE, is very generic in nature. One aspect that has remained constant throughout the past 40 years of comprehensive basic education is the approach to Lutheranism in Lutheran RE, in which instruction regarding Christianity and values is highly generic with aspects of Lutheranism often present as contextualized knowledge about Church life (Pyyiäinen 2000). Domestic research (Zilliacus 2014) has shown a contrast with so-called minority religious education curricula, which tend to be perceived as more confessional in nature, especially in the previous curriculum. The generic and relatively inclusive character of Lutheran RE may be considered to be connected to the nature of Lutheranism in the Nordic countries and to its "folk Church" status in Finnish society (NCCBE 2014, 1994). Therefore, pupils not affiliated with Lutheranism are either assumed to be so by default or are freely able to choose to attend Lutheran religious education (Finnish National Agency of Education FNAE; Ubani and Tirri 2014). However, as Ubani and Poulter (2019) indicate, the strong position of Lutheranism in RE and its weak role in public education is a more complex question that merits more investigation in its own right (pp. 203–4).

Several studies have analysed the development and role of Finnish RE in public education (Ubani 2017; Kallioniemi and Ubani 2016; Poulter 2019). The changes in Lutheran RE depict the changes

in emphasis from a tradition-centred to a humanistic pupil-centred approach (Ubani 2017). For instance, in the 1970s, Lutheran RE emphasized Bible-based generic Christian teaching (NCCBE 1970a, 1970b); in the 1980s, Christian themes were examined for the purpose of developing pupils' own personal conviction (NCCBE 1985), and from the 1990s onwards the emphasis has been on comprehensive religious general knowledge and the development of individuals' own worldview (NCCBE 1994; NCCBE 2004; NCCBE 2014; Ubani 2017). Arguably, processes of secularization have affected the subject of RE such that it has often functioned in secularist isolation, remaining largely out of step with developments of the school system, which seems, in general, to be better adapted to plurality and multiculturalism than RE (Ubani 2019).

## 4. Developments with Regard to RE Research and Diversity in Finland: The Multiculturalist Discourse

One of the issues in the background of the curriculum reform for religious education in Finland is the emphasis in recent research on religion in Finnish public education. Broadly speaking, the research on religions and worldviews in public education has been interested in matters closely related to diversification and pluralization of society and the use of topics such as RE and religion and secularity in school festivals as a means of citizenship education (Poulter 2013, 2019; Niemi 2019). For instance, several studies indicate that the role of religious traditions in schools has become increasingly complex. One question has been whether making religious objects, customs, and traditions visible in school and public spaces in general is permissible and on what principles (Poulter 2013; Niemi et al. 2014). These studies underline the negotiation between secularism and religion as one of the key issues in public education in Finland (Ubani 2019; Rissanen et al. 2019).

Arguably, in the 2000s, research on religious education in Finland has, in the main, been dominated by multiculturalist perspectives (Ubani 2017). Many Finnish researchers view multiculturalism as a context to be considered in religious education (Rissanen et al. 2019; Ubani 2013). However, there appear to be fundamental differences in opinion between researchers on how to approach this issue due to different perceptions regarding the outcome of multicultural policies and practices. From one side it has been argued that recognition of the particularity of minorities and their identities must be taken strongly into account, while the other side has called for an emphasis on assimilation in the name of secular non-confessionality and objectivity (Ubani 2013, 2019).

Research from a multiculturalist perspective has been relatively active and extensive in the 2000s in Finland (Ubani 2017). This research has tended to focus on pupils' experiences of social integration in school-wide activities (Niemi 2017), worldviews and multicultural education (Riitaoja et al. 2010), religion and multiculturalism in educational policy (Ubani 2013), school celebrations from an intercultural perspective (Niemi et al. 2014), and challenges of religious literacy and governance of religious diversity in multi-faith schools (Rissanen et al. 2020). In addition, topics such as inclusion of Muslims in Finnish schools (Rissanen 2019), supporting pupils' identities and inclusion in minority religious and secular ethics education (Zilliacus 2014), and ethical, intercultural, and interreligious sensitivities have been examined (Holm 2012).

In the context of RE, much research has recently focused on issues related to multiculturalism and on integrated RE and has become increasingly oriented towards classroom research and integrated instruction. The research has explored topics such as integrated classroom practice in RE (Åhs et al. 2019b; Korkeakoski and Ubani 2018; Ubani 2018a, 2018b), intercultural and interreligious sensitivity (Holm et al. 2019; Holm et al. 2011, 2014; Kuusisto et al. 2014), classroom dialogue and religion and conflicts in schools (Ubani 2016; Ubani et al. 2015; Kimanen 2019) and minority RE and identities (Rissanen 2014, 2019; Zilliacus 2014, 2019; Hyvärinen and Metso 2018). Recently, there has also been an upsurge in interest in research on worldviews and dialogue in religious education (e.g., Kavonius et al. 2015), and other topics such as confessionality (Kimanen 2015, 2016a, 2016b), spirituality (Tirri and Ubani 2013; Ubani 2007; Ikonen and Ubani 2014), and worldview awareness (Lamminmäki-Vartia and Kuusisto 2017) have also been studied.

Researchers have identified certain challenges regarding religious education in Finnish public school, such as a lack of opportunities for dialogue (Kallioniemi and Siitonen 2003; Kimanen 2019; Åhs et al. 2016, 2019b; Holm et al. 2019), the role of teachers in a "weak-confessional" (2007) segregated model of religious education in Finland, and the role of RE as a subject versus the role of the whole school in teaching pupils how to better understand and engage with different religions and worldviews (Ubani 2019). Lately, there have been several studies concerning integrative practices in religious or worldview education (Käpylehto 2015; Åhs et al. 2016, 2019b; Korkeakoski and Ubani 2018; Ubani 2018a, 2018b). Some researchers have gone further to suggest changing the RE model from a segregated to an integrated approach (Åhs et al. 2019a; Zilliacus 2019). Zilliacus has proposed the subject of ethics as an appropriate basis for integrated instruction on worldviews; however, most researchers view teaching on religion and worldviews as the core of the integrated subject (Ubani 2019). At the same time, some researchers point out that the current RE curriculum already offers broad opportunities for dialogue and maintain that dialogue is an integral part of pedagogical practice rather than an additional (Ubani 2019; Rissanen 2019). Overall, regardless of their differing understandings of the concept, researchers currently view dialogue as an integral part of RE and consider dialogue and encountering different worldviews as an integral part of developing citizenship and identity (Ubani et al. 2019b).

## 5. The National Curriculum of 2014 for Religious Education and Religious and Non-Religious Diversity Plurality

### 5.1. Examining Diversity in the National core Curriculum of RE

In this article, we examined the question of diversity in the national core curriculum with a primary emphasis on inductive exploration. In short, the analysis of the curriculum was collaboratively conducted via inductive content analysis. First, we read in-depth the material and identified four topics and gave them preliminary names. Then we re-read the material and identified contents for each topic. Throughout the process, we discussed as a team the reasoning behind our classifications and adapted the names of the topics to fit with the contents.

After this phase, the topics were examined deductively using Hartmann and Gerteis (2005) model of social multiculturalism. Admittedly, "multiculturalism" and "diversity" are concepts whose use is not simple. Similarl to many concepts in the social sciences, multiculturalism is an ambiguous term with several meanings, connotations, and layers: the meanings of multiculturalism range from cross-border migrants to indigenous and national minorities (see Murphy 2012). There is a broad body of literature discussing, for instance, the prevalence and understanding of "culture" in the conceptualisations of multiculturalism (Parekh 2000; Barry 2001, 2005; Gropas and Triandafyllidou 2012). The proximate concept of "diversity" has not only applied a "positive attribution" to groups beyond ethnic, cultural, or national groups, but has also broadened the sphere of discussion (ibid.).

While aware of the developments in research and literature, Hartmann and Gerteis (2005) place their work not only alongside the works of scholars such as Taylor (2001) and Alexander (2001), but also a wide range of classical sociological literature. The purpose of the model has been to lay a foundation on which to examine the "social and cultural bases for social cohesion in the context of diversity" identifiable, for instance, in literature (p. 219) on multiculturalism and diversity. Their model is based on two dimensions, of which the first focuses on the *Basis for cohesion* and the second on the *Basis for association*. The two dimensions produce four types of approaches to difference in society: assimilationism, cosmopolitanism, interactive pluralism, and fragmented pluralism (p. 224). In short, in *assimilationism*, the individuals adopt markers of the group identity of the social whole. *Cosmopolitanism* "recognizes the social value of diversity" and emphasizes individuality and individual choice (p. 228). The latter two types emphasize mediating groups more than the previous more individualistic ones. *Fragmented pluralism* views the existence of distinctive mediating communities as a positive aspect of social reality. *Interactive pluralism* acknowledges similarly the existence of distinctive groups and cultures, but emphasizes the need for common understanding, dialogue, and mutual

recognition (p. 231). As described above, these four types were used in the re-reading of the themes found in the curriculum.

*5.2. General Overview of the RE Curriculum with Regard to Diversity*

The current National Core Curriculum for Basic Education in Finland was issued in 2014 and has been in effect since 2016. The current curriculum for RE (referred to in the curriculum simply as 'Religion') aims to create common goals and core content between different religious groups. In general, "*The task of religion is to provide the pupils with an extensive general knowledge and ability regarding religion and worldviews*" (NCCBE 2014). The subject aims to promote an understanding of the relationship between religion and culture as well as multiliteracy related to religions and worldviews. In the instruction, pupils are supposed to be guided towards critical thinking and understanding of religious language (NCCBE 2014; Ojala 2017). The curriculum contains three key content areas related to the objectives of RE: (1) The pupil's relationship to his or her own religion, (2) The world of religions, and (3) The good life. The contents are selected to support the achievement of the general objectives of the subject (NCCBE 2014). The instruction of each religion is subject to these aims and in their respective syllabi the content areas are specified respectively. The overall task of the subject of religion is the same for all grades from 1 to 9, with additional task specifications for grades 1–2, 3–6, and 7–9.

In the current National Core Curriculum for Basic Education, RE aims, to some extent, to be cognizant of issues related to diversity (NCCBE 2014). This also holds true considering the previous curricula (NCCBE 1994, 2004), and aspects related to multiculturalism were also present to a degree in the previous curriculum. In terms of the key content areas, diversities are present in all three of the current national core curriculum. However, as Finnish curricula adopted skills-based approach in the latest reform, the material concerning contents in itself is scarce. The content areas are as follows: (1) The pupils' relationship to his or her own religion includes notions of diverse families and the intra-religious diversity of the religion of the pupil. (2) The world of religions mentions aspects related to cultural and religious diversity from to proximity of a school to global scale along with secularism and secular worldviews. Ecumenism and intra-religious dialogue are also mentioned here. The third key content area (The good life) includes aspects related to plurality of value positions in religions and worldviews and dialogue skills concerning religions (NCCBE 2014). When looking at the current curriculum for RE as a whole, several overlapping themes related to diversity can be identified, which we have named as "cultural diversity," "religious and non-religious worldviews," "intra-religious diversity," and "dialogue skills." These themes, explored in the following sections, show that diversity is addressed within what can roughly be called a framework of religion and multiculturality. Diversity is not represented in its full breadth of form in the Religious Education curriculum. However, "diverse families" is explicitly recognized as part of the content area "Relationship to one's own religion" (NCCBE 2014, p. 145).

*5.3. Cultural Diversity*

The first theme identified related to diversity in the curriculum can be termed *cultural diversity*. "Cultural diversity as a richness" has been defined as one of the core values of the 2014 curriculum and it is seemingly often used as a parallel or substitute term for multiculturalism in the curriculum. According to the curriculum, teaching should guide pupils "*to look at issues from the perspective of other people's life situations and circumstances*" (p. 16). Learning is considered to take place across languages, cultures, religions and beliefs. Basic education provides the basis for world citizenship that respects human rights (NCCBE 2014, p. 16).

The overarching emphasis on cultural diversity in the national core curriculum is also present in RE. According to the religion curriculum, the instruction of religion supports the pupil's growth as a responsible member of his or her community and democratic society, as well as a global citizen. Issues related to multiculturalism, such as globalization and increased international migration, are diversifying society and increasing the range of cultural views, worldviews, and religions. Religious education

is intended to support a pupil's ability to deal with a multicultural society (NCCBE 2014, p. 143.) In general, it can be stated that with regard to cultural diversity, aspects related to religion(s) and worldview(s) are the core topics of religious education. Already, the task of the subject of religion, to "*Provide pupils with extensive general knowledge and ability regarding religion and worldviews,*" includes a multicultural perspective. A general description of the subject's mission is to make multiculturalism a part of the teaching of religion at all basic school levels. The subject description states the aim of teaching pupils to consider their own and other religions and, at core, to raise awareness of different worldviews and thereby increase acceptance of multiculturalism. The instruction is intended to increase versatile knowledge of religions and to nurture respect for what others consider to be sacred (p. 143).

Cultural diversity relates to several sub-sections of the curriculum for religious education, such as objectives, contents, and arrangements. For example, for grades 1–2 (age 7–9) one of the objectives of religious education instruction is to encourage the pupil to "*familiarize himself or herself with the customs of the religions and worldviews of people in the class, at the school, and in the local area*" (p. 144). With these objectives, religious education contributes to pupils' understanding of religious and worldview diversity in the class and school environment. In grades 3–6 (age 9–13), pupils are "*encouraged in building friendships and a positive class and school community, and acting against discrimination,*" and in grades 7–9 (age 13–16) "*pupils become acquainted with the studied religion as a cultural and social phenomenon*" (pp. 265, 435). We can see multicultural education as an educational approach aimed at understanding cultures and combating discrimination in education and society (see Zilliacus et al. 2017). The RE curriculum also includes a section on guidance, differentiation, and support. In grades 1–6, the pupils' religious and worldview background and level of language proficiency are taken into account when making decisions on the activities and instruction (p. 266). The purpose of these guidelines is to ensure inclusivity regardless of cultural or linguistic background.

*5.4. Religious and Non-Religious Worldviews*

The second theme related to diversity identified in the curriculum for RE focuses on religious and non-religious worldviews. In addition to intra-religious diversity, the curriculum highlights aspects related to religious and non-religious worldviews in an inclusive manner. For instance, the aims for grades 7–9 explicitly state that pupils are to recognize the diversity of different religious beliefs. In the curriculum, religious and non-religious worldviews are discussed in the general objectives and in the contents of instruction. This theme becomes increasingly highlighted from grades 3–6 onward.

This theme is also connected to generic knowledge about religion, society, and culture, and religious literacy. The task of RE is to provide religious literacy, termed in the curriculum "Religious and Worldviews multiliteracy" (p. 265); institutional and individual aspects and secular outlooks are also included here. Finally, under this theme, religion is addressed in both an institutional sense and an individualistic worldview sense, albeit with a phenomenological emphasis. In grades 7–9, the emphasis on religious literacy becomes visibly stronger (p. 435), although this area is well represented already in grades 3–6 (pp. 264–65), and the World of Religion framework is introduced to pupils along with explicit mentions of non-religiosity (pp. 435–36).

RE is presented in grades 3–4 as a subject that provides general knowledge about religion and non-religion in society and their cultural impact in Finland and Europe (p. 264), with the context later expanded to different parts of world (p. 436). The religions mentioned in the generic objectives of grades 4–6 (p. 265) are Christianity, Judaism, and Islam due to their perceived connection to the religious and spiritual roots of Europe. This echoes the cultural heritage emphasis that was more explicitly present in the previous core curriculum for religious education (NCCBE 2004). No religions are explicitly mentioned elsewhere in the curriculum.

The contents of RE also focus on religious and non-religious worldviews and especially on identifying these in the everyday life of the pupils. For instance, in grades 3–6 in the content area "World of Religions," the current demographics and status of religions and non-religious worldviews in

Finland and Europe are explicitly mentioned (p. 265). The purpose seems to be to bring the religious and non-religious worldviews encountered in the everyday life of the pupils into the lessons, as religious communities and their places of worship are mentioned as well as non-religiosity (p. 265). In regard to learning about Finnish culture and lifestyle in the "World of Religions" content area, Christianity is the only religion or religious tradition explicitly mentioned; others are referred to as "other religions and non-religious worldviews." Interestingly, for grades 1–2 it is stated that: "In the selection of contents also non-religiosity is acknowledged' (p. 144). Finally, for grades 7–9 the aims of instruction (p. 435) mention the global significance of both religions and non-religious worldviews in decision making and gaining knowledge about their respective ethical foundations is highlighted (p. 435).

*5.5. Intra-Religious Diversity*

The third theme related to diversity that we identified in the curriculum is intra-religious diversity. The theme is highlighted in the task of the subject of religion, which states, for all grades (1–9), that: "*in teaching and learning, the pupils get familiarized with the studied religion and it's diversity*" and that "*the instruction of religion supports the pupils' ability to participate in the dialogue within and between religions and worldviews*" (pp. 143, 264, 435).

Intra-religious diversity is not explicitly mentioned in the objectives of instruction in religion or in the key content areas related to the objectives of religion. In the description of the key contents, it is mentioned in a general way that "*the pupils' experiences are taken into account in the selection of contents and in their more detailed discussion*" in all grades (1–9) (pp. 144, 265, 436). The diversity of pupils' family background is taken into account in grades 1–2 (p. 144), whereas intra-religious diversity (grades 3–6; p. 266) and themes related to it (grades 7–9; p. 436) are mentioned in the key content areas.

Intra-religious diversity is not mentioned explicitly in the objectives related to the learning environments and working methods of religion in the curriculum. However, for grades 1–2 it is mentioned that it is important that each pupil's observations and experiences are heard, and to encourage pupils to justify their opinion and accept diversity (p. 145). For grades 3–6, one of the aims is that "*the instruction respectfully reflects the diversity of religions and worldviews represented in the school*" (p. 266). For grades 7–9, it is also mentioned that "*the objective is to express the diversity of religions and worldviews in a respectful and appreciative manner*" (p. 437).

Similarly to the previous instances, with regard to guidance, differentiation, and support in religious education, intra-religious diversity is not explicitly mentioned, but it is highlighted that the pupils' different needs and backgrounds, such as language skills and cultural background, must be taken into account (in grades 1–2; p. 145). For grades 3–9, it is mentioned that the pupils' backgrounds regarding religions and worldviews and level of language proficiency are considered when implementing the syllabi and making decisions on the activities and instruction (p. 266; p. 437).

Intra-religious diversity is especially considered in the description of the syllabi for different religions. For all grades, "*the common content areas are specified in accordance with the nature of pupil's own religion*" (pp. 145, 268, 439). Syllabus descriptions are given for Evangelical Lutheran Christianity, Orthodox Christianity, Catholic Christianity, Islam, and Judaism. All of the descriptions of the syllabi for different religions in grades 1–2 take the child's own family background as a starting point for getting to know their own religion (pp. 146–48). Intra-religious diversity is approached differently with respect to Evangelical Lutheran Christianity, as can be seen from the syllabus descriptions. In the description of the syllabus for Evangelical Lutheran Christianity, the emphasis is on diversity of Christianity, whereas the descriptions of the syllabi for Orthodox Christianity, Catholic Christianity, Islam, and Judaism emphasize intra-religious diversity as viewed globally.

For example, the description of the syllabus for Evangelical Lutheran Christianity mentions that in grades 3–6, "in the selection of contents, the diversity of Christianity and, in particular, Protestantism as part of Christianity is taken into account" (p. 268), whereas the syllabus description for Orthodox Christianity states that in grades 1–2 "the pupils familiarize themselves with Orthodox traditions as a part of multicultural Orthodox Christianity" (p. 146), and in grades 7–9, "the cultural and societal

influence and visibility of religions and Orthodoxy in Finland and in countries with an Orthodox majority should be considered when selecting contents for instruction" (p. 440). The description of the syllabus for Islam mentions that in grades 7–9, pupils learn about "Islam all over the world" and that "the pupils structure their own relationship with Islam, the Finnish way of life and the Islamic world" (p. 441).

*5.6. Dialogue Skills*

Dialogue skills is the fourth theme related to diversity that we have identified in the national core curriculum for religious education (NCCBE 2014). This theme differs from the previous themes in its greater emphasis on skills development rather than acquiring content knowledge. In the curriculum of religion and its different syllabi, several levels of dialogue can be found. The concept of dialogue is mentioned only in the sense of dialogue between religions at the institutional level, although dialogue skills can be found also at the individual level when the objectives of instruction are scrutinized.

Dialogue or dialogue skills are described in the curriculum as competences that the instruction of religion should support, and are described in terms of developing competence. In the first two grades, the task of RE concerning dialogue skills is to encourage pupils to recognize and express their feelings and opinions and to learn to recognize and consider the feelings and opinions of others (p. 144). In grades 3–6, emotional and communication skills are emphasized, including supporting pupils' ability to formulate and justify their personal views (p. 265).

Dialogue skills are integral to the objectives of religious instruction for all grades. For instance, in grades 1–2 pupils are encouraged to "*recognize and express his or her thoughts and emotions,*" guided "*to be fair, empathize with other people's situations, and respect other people's thoughts and convictions,*" and "*provide the pupil with opportunities to practise expressing and justifying his or her own opinions as well as listening to and understanding the opinions of others*" (NCCBE 2014). Similarly, in grades 3–6, the objectives of instruction include creating opportunities for pupils to discuss ethical questions and to practice justifying their own views and express their thoughts and emotions constructively (p. 265). In grades 7–9, the objectives of instruction relate to argumentation skills and encountering others. Pupils are guided "*to identify and evaluate different means of argumentation and differences between religious and scientific language*" and encouraged "*to encounter different people today and in the future.*"

The key areas related to the objectives of religious instruction consist of three content areas (1) "*The pupil's relationship his or her own religion,*" (2) "*The world of religions,*" and (3) "*Good life*" (p. 144). Regarding dialogue skills, in grades 1–2 the content area is called "*good life*" and consists of empathy for other people's situation and circumstances (p. 144). Dialogue skills are also related to the "good life" content area in grades 3–6: "*pupils are provided with tools for ethical discussions and justifying their personal views as well as for discussions on religion*" (p. 266). In grades 7–9, dialogue is mentioned in the key content area "*the world of religions.*" Pupils are helped to "*understand the interaction and dialogue between religions*" (p. 436).

In the different syllabi, grades 1–2 emphasize emotional and communication skills. In grades 3–6, dialogue is mentioned as a dialogue between religions in the syllabus of Evangelical Lutheran Christianity, Orthodox Christianity, Catholic Christianity, Islam, and Judaism (p. 268–271). However, when referred to as "*a dialogue between religions*" and connected with ecumenical thinking in the syllabi of Evangelical Lutheran, Orthodox, and Catholic Christianity the view of dialogue is institutional, not individual (p. 268). In grades 3–6, pupils are to be given "*preliminary introduction to ecumenism and religious dialogue*" or "*dialogue between religions*" (p. 269). In grades 7–9, dialogue is also mentioned as a dialogue between religions in the Evangelical Lutheran, Orthodox, and Catholic syllabi, but does not appear in the syllabi of Islam or Judaism. The three Christian syllabi emphasize deepening pupils' understanding of ecumenism and the dialogue between religions and ties these to the concept of world peace (e.g., p. 439).

Finally, the objectives related to the learning environments and working methods of religious instruction in grades 1–2 emphasize the importance of hearing pupils' observations and experiences

and creating conditions for discussion, and the importance of conversation in learning. Pupils' communication and emotional skills are also mentioned. Inviting visitors to school is mentioned for all grades. The role of discussion is increasingly emphasized as pupils develop. In grades 3–6, "*discussions are emphasized in teaching and learning*" (p. 266). In grades 7–9, discussions are mentioned to be an "*important part of the instruction*" and instruction should advance "*the conceptualization of the learning topics and reflecting on the concept together*" (p. 437). The parts concerning guidance, differentiation, and support include dialogue skills. In grades 1–2, the curriculum for religious instruction emphasizes discussion and explaining key concepts (p. 145). Dialogue skills are mentioned in grades 7–9 as one of the promoted aims (p. 437).

### 5.7. Diversity in the National Core Curriculum in Light of Hartmann & Gerteis's Social Model of Multiculturalism

When scrutinised in the light of Hartmann and Gerteis (2005) typology, the Finnish RE curriculum includes aspects of different types of understanding on difference. Altogether, it seems that the ambition of the RE curriculum is to support the vision for interactive pluralism, but other kinds of visions can also be recognized (Table 1).

**Table 1.** Representation of the Hartmann and Gerteis typology in the national core curriculum for religious education (RE).

|  | Cultural Diversity | Religious and Non-Religious Worldviews | Intra-Religious Diversity | Dialogue Skills |
|---|---|---|---|---|
| Assimilationism | x |  |  | x |
| Cosmopolitanism |  |  |  |  |
| Interactive Pluralism | x | x | x | x |
| Fragmented Pluralism |  |  |  |  |

Interactive pluralism can be seen in all the themes we identified from the RE curricula. The theme of *cultural diversity*, which includes the idea that the teaching of religion supports pupils' ability to face different worldviews and contributes to pupils' understanding of religious and worldview diversity, demonstrates interactive pluralism. In interactive pluralism, it is seen that you should understand the existence of distinct groups and cultures (Hartmann and Gerteis 2005). According to the curriculum, RE helps students understand different perspectives in school and the school environment. RE aims to increase versatile knowledge of worldviews, which helps to create the ongoing interaction of different groups. This contributes to the idea of interactive pluralism. This also relates to the theme of religious and non-religious worldviews, which is connected to supporting the development of pupils' capability for religious literacy.

However, interactive pluralism was strongly represented in the theme of *intra-religious diversity* in the curriculum. The curriculum shows, for instance, that pupils are to become familiar with the studied religion and its diversity, but also pupils' ability to participate in the dialogue both between and within religions and worldviews is to be supported. The pupils' experiences and the diversity of pupils' family backgrounds are recognized in teaching. It is also notable that the learning environments and working methods of religion take note of the importance of each pupil's observations and experiences to be heard, but also encourage them to justify their own opinions and accept diversity. The objective is to express the diversity of religions and worldviews in a respectful and appreciative manner. Regarding instruction, differentiation, and support in religious education, the pupils' different needs and backgrounds, such as language skills and cultural background, are taken into account. Backgrounds in terms of religion and worldview and level of language proficiency are also to be acknowledged when implementing the syllabi and making decisions. When observing the common content areas, it can be noticed that they are specified in accordance with the nature of the pupil's own religion, and the pupil's own family background (and diversity within it) is mentioned as a starting point for getting to know their own

religion. Interactive pluralism also appears in the curriculum as an intra-religious diversity theme, and it is emphasized as globally viewed in the syllabi for minority religions: for example, learning about Orthodox traditions as a part of multicultural Orthodox Christianity, or structuring a pupil's own relationship with Islam, the Finnish way of life, and the Islamic world. In terms of Evangelical Lutheran Christianity, the emphasis is on the diversity of Christianity, and Protestantism as part of Christianity.

Even though the theme of *religious and non-religious worldviews* is somewhat presented in the light of interactive pluralism, traces of fragmented pluralism can also be seen. As described above, fragmented pluralism acknowledges the existence of a variety of distinctive communities and maintains distinctive group cultures (Hartmann and Gerteis 2005). This kind of thinking can be interpreted to be visible, for example, in the way religions are named or unnamed in the curriculum. Christianity, Judaism, and Islam are named, while the rest of the religions (or non-religious views) are referred to as "other religions and non-religious worldviews." It could be argued that this highlights the meaning of the three named religions and places unnamed worldviews in the role of the other.

The features of assimilation could be witnessed in two themes in the RE curriculum: *cultural diversity* and *dialogue skills.* Core values, cultural commitment, and minimizing distinctiveness are seen as features of assimilation (Hartmann and Gerteis 2005). One of the core values of the curriculum is "cultural diversity as enrichment" and, according to the curriculum, basic education supports a student's growth as a "responsible member of his or her community and democratic society." Arguably, if the instruction is given in accordance with the curriculum, it is always based on the values defined by the community, and it is intended to educate members for society, which can be interpreted as features of assimilation, but not exclusively.

## 6. Concluding Remarks

The purpose of this article is to examine how religious and non-religious diversity is addressed in the current core curriculum for RE (NCCBE 2014). In general, the question of religious and non-religious diversity is addressed in the context of multiculturality but there seems to be a trend toward conceptions of diversity that go beyond religion, such as family diversity. In the Nordic countries, RE is typically given tasks explicitly related to multiculturalism (Biseth 2009) and thus has broad societal and political relevance with respect to globalization, migration, and citizenship. In Finland, RE has for the past 15 years generally identified strongly with cultural diversification of society (see Ubani 2017). In the literature, it has been argued that internationally the Finnish curriculum of 2014 stands out as an example of a curriculum that strongly supports pluralism and human rights education and thus takes steps towards institutionalizing multicultural education (Zilliacus et al. 2017). Research has argued that for a number of years there has been a growing sentiment in Finland in favour of perceiving every classroom as diverse in terms of pupil gender, religion, socio-economic background, lifestyle, and values (Räsänen et al. 2018, p. 21).

In line with the findings of other general reports on the national core curriculum (Zilliacus et al. 2017), our results show that the current curriculum for RE seems to be to some extent cognizant of certain aspects of diversity. In our examination, four themes were identified with regard to addressing diversity in the RE curriculum. In general, multiculturalism and diversity were connected to religions and worldviews. The first theme identified in RE was cultural diversity. The idea given in the curriculum is that RE is to increase pupils' religious and cultural awareness and hence support their growth as broad-minded citizens. Aspects related to cultural diversity were present in religious curriculum objectives, contents, and arrangements at all basic school levels. Räsänen et al. (2018) have mentioned in their report that cultural issues are currently explicitly included in the curricula of subjects such as religion, ethics, and history.

The curriculum acknowledges the second theme identified, religious and non-religious worldviews, in several curricular areas. The third theme, intra-religious diversity, was referred to especially in the task of religious instruction and in the descriptions of the syllabi for different religions, but also in several other curricular areas. A new approach to the recognition of intra-religious diversity is being

employed in the Finnish curriculum; whereas intra-religious diversity has been previously understood to a great extent in terms of content (NCCBE 1994), the current approach also includes an element of dialogue. In addition, the recognition of non-religious worldviews, although present to a degree in the previous curriculum, is now more acknowledged in religious education.

The fourth theme identified in the curriculum was dialogue skills, which differed from the other themes in its emphasis on skills. It was also in some ways less cohesive: while the objectives of RE instruction are to support the dialogue skills of the pupils, only the institutional level of dialogue are mentioned in the key content areas of the different syllabi. Thus, the dialogue between people in everyday life seems to be absent as an element related to religion and worldviews: this finding resonates with the claim that the conceptualizations of dialogue in Finnish RE are still in a very infant phase (Ubani 2019).

The study also showed several types of understanding and objectives with regard to religious diversity, according to the Hartmann and Gerteis (2005) model. Based on the re-reading of the themes identified, the most prevalent aspect present in the curricular themes was integrative pluralism. However, here one would need to distinguish between the objectives of the curricula and the model. It could be stated that even though the RE curriculum and its different syllabi are, in general, constructed to support interactive pluralism, the model of Finnish RE is, in fact, fragmented due to the different separation of students in their respective religious education lessons.

To conclude, the Finnish core curriculum seems to adhere to several tenets issued in the international policy documents with regard to religion and beliefs in public education. The European educational policy documents such as the *Toledo Guiding Principles on Teaching about Religions and Beliefs in Public Schools* (ODIHR 2007) and *Signposts–Policy and practice for teaching about religions and non-religious world views in intercultural education* (Jackson 2014) underline the impact and possibilities that public education have for diminishing misunderstanding and negative stereotypes built on misconceptions and provocative images. In these policy documents, the acquisition of knowledge and deeper understanding about religions and beliefs is seen as a way to decrease ignorance, with ignorance seen as a route to misunderstandings and potential violence. For instance, *Signposts* highlights not only the need for knowledge but also the need for certain attitudes and skills that raise self-awareness and awareness of the beliefs and values of others (Jackson 2014, p. 21) and recommends that links are made between schools and the wider community and also with religious communities and non-religious organizations (Jackson 2014, p. 96).

As stated, several aspects presented in the aforementioned policy documents are recognized in the Finnish national core curriculum for RE to some degree. However, the recognition of diversities in the curriculum does not guarantee good practices in schools. In his works, Jackson has called for impartiality and safe space as some of the teachers' major tasks with regards to education about dialogue, religions, and worldviews in public education (Jackson 1997; Jackson and Everington 2017). However, Finnish governmental reports have recognized a lack in skills among teachers in handling religions, non-religious worldviews, and multiculturalism in schools (Räsänen et al. 2018). At the same time, some researchers have pointed out challenges in developing practices in RE, stemming partly from the alleged lack of innovative research in the field of RE (Ubani et al. 2020). Ubani et al. (2020) claim that these, along with similar trends in previous curricula, has partially contributed to a situation where the practice of RE is ill-prepared to handle questions of dialogue and integrative practices in a coherent and explicit manner (see Ubani et al. 2020; Ubani 2019). This challenge was also identifiable in the fragmented conception of dialogue in the current RE curriculum, as described earlier. In order to develop Finnish RE practice to meet the demands of today and tomorrow, the questions of dialogue, worldview inclusivity, and intra-religious diversity should be addressed in RE research, not only as practical questions, but also at a philosophical level.

**Author Contributions:** The authors have contributed both in planning and in writing of the article collaboratively. M.U. has contributed in the writing of the theoretical portion, analysis and the discussion. E.H. (Elisa Hyvärinen) has contributed in the writing of the theoretical portion, analysis and the discussion. J.L. has contributed in the

writing of the theoretical portion, analysis and the discussion. E.H. (Elina Hirvonen) has contributed in the writing of the theoretical portion, analysis and the discussion. She has carried the main responsibility over correspondence and process management. All authors have read and agreed to the published version of the manuscript.

**Funding:** This research was partially funded by the Finnish Ministry of Education and Culture as part of the project: "Creating Spaces for Diversity of Worldviews in Early Childhood Education" (2018–2021).

**Conflicts of Interest:** The authors declare no conflict of interest.

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
