# Peer review of "Dialogue, Worldview Inclusivity, and Intra-Religious Diversity: Addressing Diversity through Religious Education in the Finnish Basic Education Curriculum"

_religions, doi:10.3390/rel11110581_

Round 1

Reviewer 1 Report

This is a timely topic not only for Finland, but many other modern Western nations grappling with increased religious, social, and cultural diversity. This paper offers helpful contextual pieces to better understand how, of late, Finland approaches the intersection between religious instruction, diversity, and the education system. Moreover, this article causes the reader to think about the many aims that schools, individuals, and societies must confront when trying to learn about one’s own faith as well as other religious traditions, and then peaceably coexist with people different than one’s self. As such, the topic should be of interest to those well beyond Finland grappling with similar realities, especially since, as the authors note, Finland is highly regarded worldwide for its education system. As critical as this topic is for today’s social realities, regrettably I’m not entirely convinced that this article adds much to the larger discussions. I would offer five recommendations to strengthen the manuscript further. First, this article needs a robust theoretical framework for analysis. For example, it might make sense to draw upon a vast scholarly handling of the term, “diversity.” More than this, there is a growing and significant literature worldwide on how schools are attempting to teach (or not) on (non)religious diversity. Drawing on literature in Australia or Canada would be two such examples, among others. This literature helps to position the larger discussion in various theoretical traditions, which if applied, could yield a stronger anchor to this article. Second, the methodology is not readily clear to the reader. Can you say more about the data sources (2014 NCCBE alone?), the various elements to the sources, how the data was analyzed, and so forth. This seems like a content analysis, and thus making the method, data, and process clear would be imperative to set up the entire project. Third, this article reads at times more like a summary of what an existing report already states, and thus the reader may ask, “why don’t I just read the original report, rather than this summative piece?” What exactly is the key argument that is being advanced in this article that is different, beyond summarizing the 2014 NCCBE (as interesting as that piece truly is)? The answer to this question may dovetail with the next observation. Fourth, the four key areas that are identified do not seem to be as distinct from one another as portrayed, and moreover as acknowledged in the paper, are not even made explicit in the initial 2014 NCCBE report. I wonder about the need to go beyond the three central aims of the initial report to tease out four distinct elements in this project? For example, the opening discussion on “cultural diversity” is super helpful to frame the discussion, and seems to implicitly and explicitly get at the subsequent three ideas: learning about other religions; grappling in a reflexive way with one’s own religion and by extension the traditions of others (not sure “intra-religious” is the best term to use here; maybe “reflexive approach” or “personal reflection”?); and cultivating the skills to interact with members of different traditions. As such, I’m left wondering how truly distinct the latter three ideas are from the overarching first idea (which, again, is a really helpful idea to frame the picture – one well discussed in the literature in other national contexts). Additionally, are these four themes being projected into the initial report in ways that are not entirely necessary, if the initial three themes outlined in the report suffice? It is not entirely clear how all of these moving parts align (or not), and clarity on the earlier methodology piece may go a long way to addressing these questions. Fifth, there are several key areas in the literature that would seem plausible lines of inquiry to raise in this article. For instance: (a) How do schools assess if students are actually succeeding in the stated aims (e.g., to make friends with people of a different religion)? (b) What about teachers’ abilities to teach in these areas? Are they equipped and knowledgeable to do so, and how does the answer to this question connect with the larger discussion? (c) On the discussion of nonreligious worldviews, how exactly are these being handled? This is quite different from religious worldviews, yet there is no direct discussion of nonreligious worldviews (or literature). To reiterate, this topic is very important, and will only become more pressing as time goes on. This article offers a very solid introduction to how Finland approaches religious instruction in schools, building on an existing document/policy. If the five areas noted above are taken into account, I can certainly see ways that this article could develop in significant ways to add to a growing literature on the topic in many countries around the world. My hope is that this feedback offers some tangible and helpful steps toward that end.

Author Response

We would like to express our gratitude to the reviewers for their through work in evaluating this manuscript. We have tried to address most of the critical remarks. As most of the reviews were positive, we understood that we should stick with the character of the proposed article while reacting to the criticisms presented. Below are listed changes and comments on some decisions.

Reviewer 1:

    • “First, this article needs a robust theoretical framework for analysis.”… “Can you say more about the data sources (2014 NCCBE alone?), the various elements to the sources, how the data was analyzed, and so forth. This seems like a content analysis, and thus making the method, data, and process clear would be imperative to set up the entire project.”

It is a valid point. However, the aim of the article was to discuss the contents of the curriculum in order to highlight the main topics of article so it is not really an empirical article as such. However, we analysed the material inductively because we think as this is the first look to the topic, it is a foundation after which further analysis with a distinct framework/paradigm can be used. We added a notion of this to introduction and have emphasised the discussion function of the article.

Lines 73-77: We added this notion to 1st page: “In short, the analysis of the curriculum was collaboratively conducted inductive content analysis. First we read in-depth the material and identified 4 topics and gave them preliminary names. Then we re-read the material and identified contents to each topic. Throughout the process we discussed as a team the reasoning behind our classifications and adapted the name of the topics to fit with the contents.”

    • “…there is a growing and significant literature worldwide on how schools are attempting to teach (or not) on (non)religious diversity. Drawing on literature in Australia or Canada would be two such examples, among others. This literature helps to position the larger discussion in various theoretical traditions, which if applied, could yield a stronger anchor to this article.”

Again a good point. However, due to space and the explicit focus of special issue on Nordic contexts this is a bit of a balancing act. We have tried to refer to materials from UK (p. 10- p. 11.)which are more in the proximity of the case in Finland but expand the discussion to broader contexts.

    • “Can you say more about the data sources (2014 NCCBE alone?)” 
    • Lines 81-83: We added more information about the NCCBE2014. (p 2.)
  • “Fifth, there are several key areas in the literature that would seem plausible lines of inquiry to raise in this article. For instance: (a) How do schools assess if students are actually succeeding in the stated aims (e.g., to make friends with people of a different religion)? (b) What about teachers’ abilities to teach in these areas? Are they equipped and knowledgeable to do so, and how does the answer to this question connect with the larger discussion? (c) On the discussion of nonreligious worldviews, how exactly are these being handled?”

These are all valid question, but the curriculum in Finland in its normativity is very descriptive so it does not really give guidelines “how to” but rather “what to”. It is quite likely that in the background is the difference between German Lehrplan tradition over Curriculum in the Anglosphere: the former has emphasised the autonomous position of the teacher. It is not of course an either or question in the Finnish curriculum but this could be a distinctive aspect in relation to the US (of which we admittedly have little knowledge). So to answer the questions above: a) these are not really assessed even if part of aims, b) there have been concerns over the skills of teachers (i.e. Räsänen, Jokikokko & Lampi 2018) of which we added mentions (p 11.) and (c) There is no tradition on this yet as this is a recent development in the curriculum and there exists no empirical results nor concrete guidelines about this. It could be a question of next research, though!

Reviewer 2 Report

The article discusses how Finnish RE has addressed the issue of diversity. The authors have presented their analysis clearly and have shown why this is an important topic to address. However, I think further development of the author’s argument is needed.

Specific suggestions:

  1. It seems to me that the article has two aim: an explicit aim, and an implicit aim. The explicit aim is stated in the first line of the Introduction (line 17). It is: “The aim of the article is to discuss how religious and non-religious diversity is addressed in Finish religious education (RE).” The implicit aim is an evaluative one. The authors aim to discuss not just what is (descriptively) the case in Finish RE, they aim to discuss what Finish education should (evaluatively) be.

The implicit aim is revealed by the following passage: “the subject of RE” remains “largely out of step with developments of the school system, which seems, in general, to be better adapted to plurality and multiculturalism than RE” (lines 182-184). On a descriptive level, this passage is about RE in the school system today. On a deeper evaluative level, the passage implies that the subject of RE is deficient because it is out of step with other curricula in the school system, and that RE should be revised so that it too takes into account the plurality and multiculturalism of contemporary Finnish society.

Another example that reveals the two aims of the paper is found in lines 244-247 when the authors stated: “Overall, regardless of their differing understandings of the concept, researchers currently view dialogue as an integral part of RE and consider dialogue and encountering different worldviews as an integral part of developing citizenship and identity.” In accord with the stated aim of the article, the explicit aim, this is a descriptive statement about how researchers view dialogue in RE. However, implicit in this statement, and throughout the discussion of dialogue and RE, is the evaluative claim that dialogue should be a part of Finish RE and that changes should be made to ensure that dialogue is an integral part of Finish RE.

The implicit aim of the article is revealed most fully by the last lines of the article (509-511). The authors wrote: “In order to develop Finish RE practice and to meet the demands of today and tomorrow, the questions of dialogue, worldview inclusivity and intra-religious diversity need to be addressed in RE research, not only as practical questions, but also at a philosophical level.” I suggest that the implicit aim of the article is revealed when one notes that the authors’ intention is expressed more clearly if the word should is inserted in place of the words need to in this passage.

The author could further develop their analysis if they stated their implicit aim explicitly and reflected more fully on how they think Finish RE should be further developed today. To do so, they could begin by tweaking their opening line so that it reads: “The aims of the article are to discuss how religious and non-religious diversity has been addressed in Finish religious education (RE), and how it should be addressed as Finland is becoming a more global and religiously diverse country.”

  1. In discussing the national curriculum of 2014, the authors stated that “The curriculum contains three key content areas related to the objectives of RE: 1) The pupil’s relationship to his or her own religion, 2) The world of religions, and 3) The good life” (lines 259-261). The authors then identify and discuss four themes in the curriculum that relate to diversity. These are: cultural diversity, religious and non-religious worldviews, inter-religious diversity, and dialogue skills. (Note that the four themes are listed in the abstract in a different order than they are discussed in the article. I suggest the authors list the themes in the abstract in the same order as they are discussed in the article.) In their discussion of these four themes the authors focus primarily though not exclusively on the second objective of RE, the world of religions. That is, they focus on how a focus of the theme of diversity helps students develop a better understanding of the world of religions, including the life situations and circumstances of people of other religions in Finland today.

The authors could develop their analysis more fully by discussing more intentionally how a focus on the theme of diversity relates to the other two objectives of RE. Regarding the first objective, in order for people to address other religious and non-religious worldviews and engage in inter-religious dialogue, they need to have a secure sense of own religious beliefs. Being firmly grounded in specific religious beliefs and practices provides people with a distinctive outlook so that they have something to contribute to discussions with people of other religious outlooks and worldviews. At the same time, people need to be firmly grounded and secure in their religious beliefs and practices so that they do not feel threatened by people who express other religious perspectives, and so that they can be open to engaging in genuine dialogue with those who profess other religious and philosophical worldviews. So, in considering the RE curriculum, the authors could ask: to what extent and how does the existing curriculum help a pupil develop a secure relationship to his or her own religion and be able, from the perspective of this religion, to be non-anxiously open to dialogue with people of other religions? Similarly, with regard to the third objective, the author could, in examining the curriculum, ask: To what extent does the curriculum enable educators to teach pupils to identify common understandings of the good life and to address differences in what it means to life a good life among people of diverse cultural and religious backgrounds?

  1. In lines 172-173 the author wrote: “its weak role in public education a more complex question.” I think this should be: “its weak role in public education is a more complex question.”

  1. The authors might find it helpful to expand the resources from which they draw insight. For instance, they mention but do not delve into the work of religious educator Robert Jackson. Jackson and other British religious educators have been addressing issues of religious diversity for many years.

Author Response

Reviewer 2:

  1. ” The authors might find it helpful to expand the resources from which they draw insight. For instance, they mention but do not delve into the work of religious educator Robert Jackson. Jackson and other British religious educators have been addressing issues of religious diversity for many years.”

  1. 11. Lines 534-541. We have increased the use of the work of Jackson in the discussion.

  • Changes made:
    • Lines 509-511: we took note of the amendment and changed word need to proposed word should
    • Lines 172-173: we took note of the amendment and added the proposed word is

Reviewer 3 Report

Dear Authors

Although your research is informative to understand the background and the current status of the national core curriculum for religious education, on diversity, in Finland, it doesn't go beyond a simple presentation of factual matters.

Introduction does not provide a clear roadmap for your argument. It does talk about the problem of "integrated instruction." But, it does not articulate why this matter is significant and how you will address it in this paper for what objective. The multiculturalist approach, which is the anchor of your larger arguments, needs to be more articulated in the introduction than it is.

However, what is more problematic is your excessive focus on the simple interpretation of the 2014 National Core Curriculum for Basic Education on "diversity." The interpretation covers four themes: cultural diversity, religion and non-religious worldviews, intra-religious diversity, and dialogue skills. Unfortunately, you provide superficial interpretations for each theme.There is no critical analysis, comparison, or reflection. Nor does the discussion offer any argument of why your discussion is significant in advancing the already established scholarship of religious education in Finland.

I recommend that you develop and add your critical analysis and reflection to show an originality of your research and more actively engage with the established literature to highlight your contribution. 

Author Response

Reviewer 3:

  1. “It seems to me that the article has two aim: an explicit aim, and an implicit aim. The explicit aim is stated in the first line of the Introduction (line 17). It is: “The aim of the article is to discuss how religious and non-religious diversity is addressed in Finish religious education (RE).” The implicit aim is an evaluative one. The authors aim to discuss not just what is (descriptively) the case in Finish RE, they aim to discuss what Finish education should (evaluatively) be.”
  1. The author could further develop their analysis if they stated their implicit aim explicitly and reflected more fully on how they think Finish RE should be further developed today. To do so, they could begin by tweaking their opening line so that it reads: “The aims of the article are to discuss how religious and non-religious diversity has been addressed in Finish religious education (RE), and how it should be addressed as Finland is becoming a more global and religiously diverse country.

  • P 1. Lines 26-29. We added an explicit reference of our position with regards to diversities so that the underlying position of the authors and article recognized (rightfully) by the reviewer is explicitly given out so that the reader is aware of the normative value position of the authors. As noted in the review, we return to this topic again in the discussion. 
  •  
  1. In discussing the national curriculum of 2014, the authors stated that “The curriculum contains three key content areas related to the objectives of RE: 1) The pupil’s relationship to his or her own religion, 2) The world of religions, and 3) The good life” (lines 259-261). The authors then identify and discuss four themes in the curriculum that relate to diversity. These are: cultural diversity, religious and non-religious worldviews, inter-religious diversity, and dialogue skills. (Note that the four themes are listed in the abstract in a different order than they are discussed in the article. I suggest the authors list the themes in the abstract in the same order as they are discussed in the article.) In their discussion of these four themes the authors focus primarily though not exclusively on the second objective of RE, the world of religions. That is, they focus on how a focus of the theme of diversity helps students develop a better understanding of the world of religions, including the life situations and circumstances of people of other religions in Finland today.

  • P 10. Lines 290-299. Here is added an explanation of the curriculum: as Finland adopted a skills-based approach in the curriculum, the content descriptions are very limited. But to meet with this critical comment, we added a description what could be found from the material, but in our views it is not sufficient for more in-depth analysis.
  •  

Reviewer 4 Report

This is a well-written highly informative article. I have some questions and comments.

P2l50-51 To what sector/grades does this refer?

P2l88 Grades?

P3l137 Do the pupils get information on other religions?

P4l155-160 How many inhabitants has Finland?

P4l174 How is the Finnish educational system structured? Are there in addition to public schools also private schools; are these schools denominational schools; if so, how many denominational schools exist per religious community?

Is RE a separate subject; are there special RE teachers; if so, do they have specific qualifications/training; are there RE examinations; who has written the RE syllabi; are there structural differences between the various syllabi?

Author Response

Reviewer 4:

  • Changes made:
    • P2150-51: we took note of the amendment and corrected elementary schools
    • P2188: we added clarification to the introduction (see line 19): The Finnish basic education consists grades 1 to 9 and covers ages 7 to 16.
    • P31137: we added clarification to the introduction (see lines 139-140): information on other religions and worldviews
    • P4155-160: we took note of the amendment and added sentence (see line 151): Finland has 5,53 million inhabitants.
    • P41174: we took note of the amendment and added to the introduction (see line ___) the sentence: …the Finnish children attend to public schooling. There are almost 2300 elementary schools (OSF 2018). Under 100 of those are private schools and under 20 schools are with some religious devotion.
    • Question about teachers qualifications answered in the introduction as follows: At grades 1 – 6 RE is teached by class teachers and at grades 7 – 9 teaching is provided by subject teachers specialized with religious education.

Round 2

Reviewer 3 Report

Dear Authors

Since your response to reviewers' comments and suggestions does not address my concerns, my previous decision still stands. Your presentation of the 2014 NCCBE and your articulation on the new emphasis suffer from a lack of depth. As I noted in my first review report, there is no critical reflection, analysis, or comparison. It seems that your main body argument is simply to identify, highlight, and reiterate, with a little detail, the wordings of the NCCBE2014 on "diversity" and "dialogue." I don't think that this research in its present form would make any meaningful scholarly contribution to our understanding of RE in Finland. 

Author Response

Dear Reviewer, 

We have now tried to accommodate the requests & criticisms with regards to diversity and analysis in the article. In practice we placed the portion with regarding to diversity right before the four original themes of analysis and then used the Hartmann & Gerteis typology in the re-reading of the themes. This decision was made so that we would not break the flow of the theoretical portion and so that we would not make it longer than it already is. Now the typology is also nearer to its use when we re-read the themes in light of these themes. The new portions are in red so the changes can be examined swiftly.

Round 3

Reviewer 3 Report

Dear Authors

I think that your utilization of Hartmann & Gerteis typology does enhance the originality of your research and your critical approach to NCCBE2014. For my last question, is putting periods in the manuscript title intentional or just a technical error? It needs to be corrected. The English of this paper needs to be improved. 

Author Response

Thank you for your review. The last comments are noticed and corrections made. You can find them in the corrected version of the paper.